# A Review of Structural Adhesive Joints in Hybrid Joining Processes

**DOI:** 10.3390/polym13223961

**Published:** 2021-11-16

**Authors:** Sofia Maggiore, Mariana D. Banea, Paola Stagnaro, Giorgio Luciano

**Affiliations:** 1Dipartimento di Chimica e Chimica Industriale (DCCI), University of Genova, Via Dodecaneso 31, 16146 Genova, Italy; sofia.maggiore@edu.unige.it; 2Istituto di Scienze e Tecnologie Chimiche “Giulio Natta”—CNR, Via De Marini 16, 16149 Genova, Italy; paola.stagnaro@scitec.cnr.it; 3Federal Center of Technological Education of Rio de Janeiro (CEFET/RJ), 20271-204 Rio de Janeiro, Brazil; mdbanea@gmail.com

**Keywords:** epoxy adhesives, polyurethane adhesives, acrylic and cyanoacrylate adhesives, adhesive joints, hybrid joining technologies (HJ), hybrid bonded–bolted (HBB) joints, resistance spot welding and adhesive bonding (RSW-AB), friction stir welding and adhesive (FSW-AB)

## Abstract

Hybrid joining (HJ) is the combination of two or more joining techniques to produce joints with enhanced properties in comparison to those obtained from their parent techniques. Their adoption is widespread (metal to metal joint, composite to composite and composite to metal) and is present in a vast range of applications including all industrial sectors, from automotive to aerospace, including naval, construction, mechanical and utilities. The objective of this literature review is to summarize the existing research on hybrid joining processes incorporating structural adhesives highlighting their field of application and to present the recent development in this field. To achieve this goal, the first part presents an introduction on the main class of adhesives, subdivided by their chemical nature (epoxy, polyurethane, acrylic and cyanoacrylate, anaerobic and high-temperature adhesives) The second part describes the most commonly used Hybrid Joining (HJ) techniques (mechanical fastening and adhesive bonding, welding processes and adhesive bonding) The third part of the review is about the application of adhesives in dependence of performance, advantage and disadvantage in the hybrid joining processes. Finally, conclusions and an outlook on critical challenges, future perspectives and research activities are summarized. It was concluded that the use of hybrid joining technology could be considered as a potential solution in various industries, in order to reduce the mass as well as the manufacturing cost.

## 1. Introduction

The development of efficient and flexible joining methods for structures is a critical step for cost-effective manufacturing, as well as for the repair and replacement of components in various types of structures in industry. The main joining technologies currently used in industry are: mechanical fastening, welding and adhesive bonding.

Adhesive bonding provides several advantages for joining metals when compared to resistance spot welding or mechanical fasteners, such as rivets or screws [1]. It is the technique of choice for bonding dissimilar metals, creating bonds at low temperature, combining bonding and sealing in one operation, providing thermal and electrical insulation, distributing stress uniformly, producing a smooth surface appearance and providing fatigue, vibration and sound damping. It saves weight (in many applications) simplifying the design of the structure [2,3]. Moreover, although it has several advantages compared to other joining technologies, it is more labor intensive due to surface preparation steps and long curing times. Adhesive bonding limitations are related to peel strength, operational temperature limit, delayed completion because of curing time, testing procedures and restricted service conditions [2]. Issues related to its strength durability and environmental degradation actually limit its use on a large scale by the industry [4].

The decisions whether to use adhesives, mechanical fasteners, welding, or a combination of these methods depends on various factors. It must be remarked that all the different joining processes are not generally competitive between them and should rather be considered as being complementary. The appropriate joining technology for any application should be chosen between the best trade-off technological and/or economic value.

In hybrid joining technologies, two or more joining processes are carried out, combining the best characteristics of each parent process. The main advantages of combining an adhesive with another joining method are: improved fatigue resistance (static and dynamic), more uniform stress distribution, peel and impact resistance (the point joint arrests crack growth in the adhesive bond), reduction in structural weight, better load sharing, sealing properties, production of continuous joints and increased safety.

The most commonly used adhesives in hybrid joining processes are the structural ones which are selected with the aim of fastening together elements in order to produce high modulus, high strength and permanent bonds. They must be capable of transmitting structural stress without loss of structural integrity within design limits [5].

### 1.1. Epoxy Adhesives

Epoxies can be formulated to meet a wide variety of bonding requirements. Systems can be designed to perform satisfactorily at low (e.g., −150 °C) and high temperature (e.g., 200 °C). Epoxies yield good to excellent bonds to steel, aluminum, copper and most other metals and they have a wide field of applications. Epoxy-based adhesives offer the advantages of relatively low cure temperatures, relatively low cost and a variety of formulating and application possibilities. They also have the advantage that no volatiles are released during the curing process.

### 1.2. Polyurethane Adhesives

Polyurethanes (PU), whether one-part or two-part, are high-performance and very versatile and have a wide range of properties, including adhering to a varied range of substrates. Generally, the *T*_g_ of polyurethanes is higher than that of silicones, but lower than that of highly cross-linked, structural epoxy resin adhesives [6,7]. The outstanding features of polyurethanes are good mechanical properties at low temperatures (i.e., high elongation capacity, high energy absorption capacity, high resistance in aggressive environments, thermal stability and chemical resistance) [8].

### 1.3. Acrylic and Cyanoacrylate Adhesives

Acrylic adhesives polymerize by a free-radical addition process. They present excellent adhesion to unprepared or minimally surface prepared metals, composites and most thermoplastics.

Cyanoacrylates cure through reaction with the alkalinity, in the form of moisture, held on the surfaces to be bonded. Cyanoacrylates are thermoplastic when cured and are limited in temperature capability and chemical resistance. Cyanoacrylates provide high bond strengths on plastics and rubber materials. Bond durability problems are encountered with silicones, polyolefins and some fluoroelastomers and with glass, where extensive surface preparation is required. In general, cyanoacrylates have an excellent lap-shear strength and good shelf life [9].

### 1.4. Anaerobic Adhesives

Anaerobic adhesives are one-part acrylics that remain monomeric in the presence of air but polymerize in its absence. They are supplied in air-permeable containers. Since polymerization depends on the exclusion of air, anaerobic adhesives are best suited to the bonding of machined components. They have good resistance to solvents, and for these two reasons are used in automotive engine construction: for example, in sealing between castings and in the retention of gear trains on their shafts. In aircraft they are used for fixing roller bearings in instruments [9].

### 1.5. High-Temperature Adhesives (HTAs)

HTAs based on bismaleimides and polyimides are available for service temperatures up to 290 °C. They are cured under pressure at 175–200 °C and provide useful strength 15 MN m^−2^ in the lap-shear test) up to the maximum allowable temperature. Such adhesives are expensive and not for general use [10].

The adhesive selection process is difficult as there is no universal adhesive that will fulfil every application and selection of the proper adhesive is often complicated by the wide variety of available options [11]. In general, in order to better organize the selection of the most suitable adhesive for a candidate application, the following factors should be considered: the type and nature of substrates to be bonded, the cure and adhesive application method, the expected environments of service, and the related stresses that the joint will be subjected to in service. The expected end-use environment is perhaps the most important of these considerations. The environmental concerns, such as temperature changes, chemical and UV exposure, electrical insulation and other important factors, are considered for best bonding strength [12]. In general, the basic requirements of the adhesive used in hybrid joining are the same as those for adhesives used in conventional adhesive bonding processes. However, they must have some specific characteristics depending on the hybrid process used. For example, for weld-bonding processes, the adhesive should have sufficient heat resistance to the welding temperatures so as not to adversely affect the strength of the final hybrid joint. One-part heat-curing adhesives are usually used in hybrid joining processes used in automotive industry, avoiding the need for clamping by using the point joints to hold the bonded parts together and employing the baking process to cure the adhesive [13].

There are many recent review articles regarding joining techniques of same and dissimilar materials combining mechanical fastening, welding and the use of adhesives. The focus of these reviews is to describe the process of joining and their mechanical and physical properties via simulation and laboratory experiments [1,14,15,16,17,18,19,20,21,22]. To the authors’ knowledge, one topic that was not covered in significant detail is related to the use of adhesive in hybrid joints (combination of adhesive bonding with mechanical joining and welding processes). For this reason, in this work, an overview of recent advances on hybrid joining processes are discussed following structure (see Figure 1):Introduction of the main class of adhesives, subdivided by their chemical nature considering epoxy, polyurethane, acrylic and cyanoacrylate, anaerobic and high-temperature adhesives;Description of the most commonly used HJ techniques (mechanical fastening and adhesive bonding, welding processes and adhesive bonding);Recent advances on the application of structural adhesive joints in hybrid joining processes.

## 2. Hybrid Joining Processes

Hybrid joining processes combine fundamental processes to achieve some unique characteristics or capabilities, often with synergistic results. These hybrid processes tend to be of more recent origin and have often been developed to overcome shortcomings of the basic processes for particularly challenging applications (e.g., advanced aerospace structures). As a result of the progress in the field of material engineering, development works are orientated to using thin sheets and small constructional elements. There is also a growing trend of combining traditional metals with polymeric composites. For instance, composites are more structurally efficient, in terms of strength-to-mass ratio, than metals and do not experience galvanic corrosion, while metals have better damage tolerance and failure predictability than composites and are unaffected by the solvents and temperatures which tend to degrade polymers. Therefore, in order to optimize the benefits provided by both types of materials, multi-material joints between metals and composite materials are increasingly being developed [16]. To bond this type of constructional components and the polymer–metal multi-material structures special joining techniques are needed. Among those the most used are hybrid processes of the main processes of mechanical fastening, adhesive bonding and welding. In the next subsections, the main types of hybrid joining processes that use structural adhesives are briefly presented.

### 2.1. Advanced Fastening and Bonding Processes

The main objective behind the developments in advanced fastening and bonding processes is to successfully join new advanced materials with excellent joint properties. The principal techniques of advanced fastening and bonding methods are named hybrid bonded fastened joining and clinch bonding (clinch + adhesive). Hybrid-bonded fastened joints and clinch bonding are produced by simultaneous actions of adhesive bonding and a mechanical fastening technique.

#### 2.1.1. Hybrid-Bonded Fastened Joints

A hybrid approach of bonded fastened joints is developed in order to avoid individual limitations of fastening and adhesive joining for assembling of dissimilar materials, composites or composite materials and metallic materials. There are two types of hybrid-bonded fastened joints, such as the joints that use bolts/rivets known as hybrid-bonded–bolted (HBB) joints (see Figure 2a,b adapted from [23]), and the joints that use pins with adhesive, known as hybrid-bonded pinned joints (see Figure 2c).

A fastening system can help to sustain axial loads while the adhesive takes some of the load off the fasteners and redistributes the remaining load more uniformly. The fasteners, in turn, also take some load off the adhesive, particularly out of plane. Most important parameters such as type of joint configuration, type of fastener, adhesive material and its thickness (bond line), loading condition, type of adherend being used and its thickness, overlap length and fastener–hole clearance of hybrid bonded fastened joints are required to produce metallurgically sound joints [24,25]. Table 1 shows process parameters reported under the literature of hybrid bonded fastened joints.

#### 2.1.2. Clinch Bonding

Clinching is known as a press-joining technique, aimed to join thin sheets using specially designed tools or without application of the tool, through interlock formed by plastic deformation of base materials. Clinching is an alternative joining method of riveting, screwing, adhesive joining and spot welding [27,28,29,30]. Clinching can be applied to join metal to metal, for joining of aerospace materials such as composites, aluminum, titanium, composite metal dissimilar joints, and for manufacturing applications of ropeways and different nautical equipment [31].

Clinch with adhesives is called clinch bonding, where suitable adhesive is applied between workpiece sheets. Figure 3 illustrates the technological process of clinch bonding. It is a modern and innovative technology allowing connection of different types of materials to create durable and reliable light constructions. However, its practical implementation is still very limited. In comparison to a single joint, the application of clinching together with adhesive bonding leads to an improvement in: the quality, rigidity and the load capacity, dumping of noise and vibration, pressure tightness and corrosion protection [31]. Compared to other mechanical joining techniques the clinch-bonding technique has the advantage of employing only the adhesive as consumable. The adhesive is applied to one of the parts being joined, and after that the two parts are placed together. The parts being joined are subjected immediately to the clinching process, and this process can cause some adhesive to ooze out of the joint. Finally, once clinching has taken place, the joint is left to harden.

### 2.2. Welding Processes and Adhesive Bonding

Hybrid techniques of adhesive bonding and welding processes can be subdivided as weld bonding and other hybrid welding processes. The first is a hybrid of resistance spot welding and adhesive bonding (RSW-AB), while the second includes Friction Stir Welding (FSW-AB), Friction stir spot welding (FSSW-AB), Laser Weld Bonding (LBW) and Laser Spot Weld Bonding (LSWB). Typical features of RSW-AB and FSW-AB are briefly presented in the next subsections.

#### 2.2.1. Resistance Spot Welding and Adhesive Bonding (RSW-AB)

Traditional techniques—and, particularly, welding—are not always easy to abandon, because of reliability, widespread knowledge and controllability of the process. This is the case of resistance spot welding (RSW), which is a well-known joining technique still widely employed today for assembly of body structural parts of vehicles. Figure 4 presents a schematic illustration of a joint produced by a hybrid weld-bonding joining process. In this process, a layer of adhesive, in either paste or film form, is applied to one of the metals to be joined. The other metal member is placed on top, forming a lap-type joint, and the assembly is then clamped to maintain part alignment. The two metal members are then joined by resistance welding through the adhesive using a spot welder mounted on a common C-frame, or as a portable unit attached to the working end of a robot arm [33]. Spot welding pressure and heat displace the adhesive and allow metal fusion to form a nugget. A visible mark on the face surface of the metal sandwich sheet denotes the weld location.

In the last few years, the process has gained importance in the aerospace, automotive and railway industries because it combines the advantages of spot welding, in terms of mechanical and thermal strength, with the stress concentration softening and tolerance to structural damage that are typical of adhesives [34]. There is evidence that static and especially fatigue strength of the weld-bonded joints are better than the purely added strengths of the adhesive and the spot welds. This probably has to do with spreading the loading or stress distribution through the adhesive and minimizing stress concentrations [35]. Additionally, weld-bonded joints offer increased static tensile and/or shear strength due to the increase in the total area of joining given by the area of the welds and the adhesive. Another key feature of the weld-bonding technology is the possibility of replacing discontinuous spots by continuous joints, thereby increasing the fatigue resistance and impact resistance of the vehicles, optimizing the quality of the seal and improving the overall resistance to corrosion. The technology of hybrid adhesive-welded joints may be applied to link up steel components (including stainless and coated steels), aluminum or titanium alloys and even composite materials [36].

#### 2.2.2. Friction Stir Welding and Adhesive Bonding (FSW-AB)

As previously mentioned, Friction Stir Welding (FSW) is the one of the most promising processes in continuum joining, and although the FSW has several advantages over other welding techniques as regards joining aluminum alloys, it also has counter-difficulties. An example of the challenges that friction stir welding brings is, in the case of overlap configuration joints, the presence of a hook defect reduces the static and fatigue strength as this defect acts like a crack initiation point. Furthermore, the question of chemical corrosion of certain alloys is also a factor requiring prevention by the use of sealants [37,38,39,40,41]. Therefore, the use of a hybrid joining technology, which combines the FSW and adhesive bonding (AB) forming friction stir weld-bonding (FSW-AB), could present itself as a solution for these concerns. The development of this new joining technology aims at incorporating properties and characteristics of both joining technologies, as well as improving damage tolerance. FSW can produce consistent joints with high static strength, while the adhesive will not only allow improved vibration damping and fatigue strength but may also serve double duty as a sealant, isolating the weld from the environment. Figure 5 shows a schematic illustration of the FSW-AB process.

The hybrid joining technology FSW-AB presents different challenges, such as:The process’ temperature range is between 300 and 470 °C [41];The chemical nature of the polymeric adhesives used is relevant, in particular with reference to their thermal resistance (during the process) and chemical resistance (during the use in corrosion environment);The avoidance as much as possible of solvents, or nonenvironmentally friendly additives, in order to reduce the environmental impact of the process.

The FSW also finds its application in the field of welding of wood as in the works of Gfeller et al., Kanazawa et al. and Leban et al., where the combination of friction welding with the melting of wood lignin functioning as molten matrix of entangled fibers due to the friction welding action is studied [42,43,44]. FSW is used in the welding of wood plastic composites as reported by Rahbarpour [45] in order to avoid mechanical fasting methods and problems related to stress concentration, galvanic corrosion and mismatch in the coefficient of thermal expansion and damage of reinforcing fibers induced by drilling. Recently (2020) Xie also reported an application of Friction Stir Spot welding for joining aluminum and woods using an intermediate polymer layer conferring superior tensile shear strength and overcoming the issues of inducing chemical bonding, insulating the heat input via a polymer layer, increasing the interlocking and surface roughness and to avoid the degradation of wood [46].

## 3. Application of Structural Adhesive Joints in Hybrid Joining Processes

Recent advances on the application of structural adhesive joints in hybrid joining processes are presented.

### 3.1. Fasteners and Adhesive Joint

In the following sections the application of structural adhesives is described in dependence of the mechanical fastener used.

#### 3.1.1. Hybrid Bonded Fastened Joints

The HBB joints with multiple fastener joints have been studied experimentally and numerically by numerous authors [47,48,49,50,51,52,53,54,55,56,57,58,59,60,61,62,63,64,65,66,67,68,69,70,71,72,73,74,75,76,77,78,79,80,81]. A study by Chowdhury [47] demonstrated that the presence of the bolts in practice could provide the HBB joints with a fail-safe mechanism which bonded joints alone are not able to offer. Further confirming the damage tolerance of HBB joints, Fu and Mallick [48] studied fatigue testing and quantified the superior fatigue life of hybrid joints compared to bonded joints. While this research work showed the feasibility of utilizing adhesive bonding in aerospace applications, the HBB joints were still considered an inefficient design because the adhesives were taking the majority of the load through the joints. This was due to the use of conventional adhesives whose stiffness was too high for effective load sharing between the bond and the fasteners [49,50,51]. The realization of the unbalanced load sharing led to the use of flexible adhesives in HBB joints. Kelly [49] predicted the load sharing observed two types of HBB joints, one with stiff epoxy adhesive and another with flexible polyurethane adhesive. His experimental study showed that in the case of a stiff adhesive layer that load transmitted by the bolt is little; therefore, its contribution to the improvement of simple bonded joints is limited. By contrast, the effect is greater if either the adhesive layer or the adherends are more compliant, which means lower elastic modulus, higher bond line thickness and lower adherend thickness. Analytical models were developed to evaluate the stress [52], predict load transfer in bolts and adhesive joints and predict the strengths of the bolted–bonded composite joints [53].

The improvement of HBB joints’ performance compared to both individual joints is achievable when there is substantial load sharing between the adhesive and the bolts. The load sharing in the HBB joints depends on a combination of materials and geometric parameters and attaining an appropriate load distribution in the joint design is not a trivial task. For example, Lopez-Cruz et al. [54] demonstrated the higher strength of HBB joints with flexible adhesive, compared to those with stiff adhesive. The improved joint strength was due not only to the effective load sharing, but also to the ability of the flexible adhesives to relieve stress concentration in the joints’ bond line [55,56]. Consequently, a more exhaustive understanding on flexible adhesives’ properties follows in order to advance HBB joint design. For example, with the experiments of Lopez-Cruz the adhesive stiffness was found to be highly relevant to the joint strength [54]. Crocker et al. [57] characterized flexible polyurethane adhesives through uniaxial, equibiaxial, and planar tension tests then modeled them with hyper-elasticity. Duncan and Dean [58] characterized a rubber-toughened epoxy adhesive, and numerically simulated bonded joints using a hyper-elastic model. Kelly [49] tested the polyurethane adhesive under tension and used an elastic/plastic model to perform finite element analysis. Bodjona and Lessard [24] conducted a global sensitivity analysis using a variance-based statistical method to study the effect of the different joint design parameters on the load sharing and reported that the adhesive yield strength is the most important factor influencing the load sharing. Several studies that used strong and stiff adhesives (i.e., FM300, FM73, Hysol EA 9317) and thin adhesive thickness failed in enhancement of the hybrid joint strength compared to the strongest parent joint [59,60,61].

More recently, Liu et al. [62] conducted an experimental investigation of the load-bearing behavior, and ductility of FRP hybrid double-lap joints composed of both adhesively bonded and bolted connection parts was conducted. The effects of the adherends’ fiber architecture (uni- or multi-directional), adhesive type (stiff or flexible), and displacement rate were examined. The authors showed that the resistance of the hybrid joints with flexible adhesive corresponded to the full summation of the resistance of the bonded and bolted connection parts. The ultimate failure loads of these joints were significantly improved by increasing the displacement rate, while the deformation capacity did not decrease. Finally, their experimental results showed that the hybrid joints with multi-directional adherends and flexible adhesive exhibited high joint efficiencies and excellent ductility. Zhang et al. [63], systematically studied the failure behaviors of the hybrid joint (HJ) of plain-woven carbon fiber-reinforced plastic (CFRP) and aluminum alloy. The HJs were compared with bonded joints and riveted joints through experiments, and failure behaviors of the HJs were discussed. The adhesive materials used were a polyurethane and epoxy adhesive. Experimental load-displacement curves of HJs, bonded joints and riveted joints are shown in Figure 6. In Figure 6a, the curves of HJs with six different bond line thicknesses show obvious regular distribution characteristics. As the thickness of the bond line increases, the initial rigidity and peak load of the HJs decrease. When the displacement exceeds ~1.0 mm, the curves return to approximately linear until the maximum load position. The reason for this phenomenon can be found in Figure 6c. For bonded joints, the rigidity is always approximately linear. While for riveted joints, when the loading displacement is in ~0.5–1 mm, damage occurs around the holes of CFRP and the curves are obviously nonlinear. Therefore, for HJs, the nonlinear interval is caused by the damage of rivets and CFRP. It is worth noting that the curves of HJs with different bond line thicknesses fall back from the peak load to ~5 kN, and this phenomenon shows that when the bond line is completely broken, the HJs become approximately equivalent to riveted joints. The failure processes of the HJs can be subdivided in three stages (as shown in Figure 6c): first, the bond line and the rivet share the load; then the bond line gradually fractures; finally, the rivet bears the load alone. The authors concluded that HJs are superior to bonded joints and riveted joints in strength and energy absorption.

Imanaka et al. [64] investigated the combination of blind rivets and adhesive bonding in lap-joints made from high strength steel. The effect of joint width and the adhesive properties on the fatigue life were investigated. The fatigue life of bonded and riveted/bonded joints with an acrylic adhesive were found to be approximately equal but increased fatigue life was observed in the hybrid joints with an epoxy adhesive. The fatigue strength of the joints increased with reduction in joint width to hole diameter ratio. The propagation rate of fatigue cracks was reduced in riveted/bonded joints in comparison with adhesive bonded joints. As reported by Habibi [65] et al. the adhesive–rivet joint also provides the possibility of simultaneously having the strength of the rivet joint along with a more uniform distribution of stress related to the adhesive joint. Gómez et al. [66] proposed a simple analytical model combining springs and dampers, which could reproduce the behavior of a structural adhesive/riveted hybrid single lap joint with less than 15% error level compared with experimental curves. Simplified finite element models were also proposed to analyze stress distribution and further predicted the joint performance. 

The influences of different manufacturing processes, substrates and rivet materials on structural performance of riveted joints were studied in [49,50,67,68,69]. For example, Li et al. [70,71,72] compared the pros and cons of the electromagnetic riveting process against the traditional riveting process, and they revealed that the former was more stable than the latter, and the rivet experienced shear failure. Di Franco et al. [73] explored the effects of the self-piercing riveting (SPR); they found that self-piercing riveting could damage the continuity of fibers, thereby reducing the mechanical properties of CFRP substrates. In these studies, inserting direction of rivets was not taken into account. In riveting joints, through-holes in substrates can considerably degrade joint strength as a result of stress concentration or delamination of composite substrates [32]. 

The mechanical behavior of hybrid rivet-bonded joints was also studied by Pirondi and Moroni [26,74]. They carried out a vast experimental study to compare simple and hybrid joints in terms of strength, stiffness and energy absorption, also accounting for the influence of some geometric and environmental variables. The authors found that hybrid joints (rivet-bonded joints) increase the maximum load and the initial stiffness compared to a joint with a rivet alone.

Marannano et al. [75], in 2015, carried out a numerical–experimental study of bonded/riveted double-lap joints between aluminum and carbon fiber-reinforced polymer (CFRP) laminates. They permitted to highlight both the static and the fatigue performance of such joints obtained by using aluminum and steel rivets. Figure 7a,b shows the average values of the tensile strength and the stiffness of the different junctions examined. The experimental data were obtained by tensile tests on simple adhesively bonded joints (SI), simple riveted joints (SR) with aluminum (SR_AL) and steel (SR_STEEL) rivets and hybrid joints with aluminum (HYBRID_AL) and steel rivets (HYBRID_STEEL). The authors demonstrate that the introduction of steel or aluminum rivets on an adhesively bonded joint with an overlap length equal to twice the minimum size recommended by the theory allows improvement the static strength and, especially, the fatigue performances. 

Sadowski and Golewski [76,77,78], used 13 different models of hybrid joints with the same adhesive area to study the influence of shape on joint strength. They analyzed different distributions of fasteners for the single and double-lap hybrid joints. It was found that the initial stiffness of the joints was not affected by the chamfer size, but the use of chamfers significantly increased the strength. They state that the fasteners should be kept away from the lap axis perpendicular to the direction of tension force in order to achieve optimal distribution for multi-fasteners.

Presse et al. [79] investigated the fatigue life of multi-material connections hybrid joined by self-piercing rivets (SPR) and adhesive. The fatigue life estimation of SPR and of the adhesively bonded connections was based on a structural stress approach and material SN curves. Both numerical assessment concepts were developed on various material and thickness combinations and show a good agreement with the experimentally observed fatigue life.

Romanov et al. [80] performed a parametric study on static behavior and load sharing of the multi-bolt HBB composite joints. Five different geometry configurations were studied: the overlap length, distance between the bolts and edge distance to the bolts. Their studies have shown that the load sharing between the adhesive and the bolts is shown to be geometry dependent, i.e., facilitated by a shorter joint overlap length and smaller bolt-edge distance. The overlap area is shown to be a dominant factor for the strength improvement over that of the load sharing. However, providing that the overlap area is kept unchanged, enhanced load sharing leads to a higher joint strength, revealing their nonlinear relationship.

Selahi [81] employed a simulation software to perform failure analysis of hybrid (bonded and bolted) single and double-lap joints. His analysis manifested that the HHB joint are more appropriate for single-lap form, and double-lap should be avoided, but more research is needed to confirm whether this is a universal rule.

#### 3.1.2. Clinch Bonding

Relatively fewer investigations have been made on the clinch-bonding hybrid technique because its practical implementation is still very limited in manufacturing industry [82]. The following are the limited few studies found in the literature. Gerstmann et al. [83] described the development of a special type of clinching, namely flat-clinch bonding that combines flat clinching and adhesive bonding, using numerical simulations. In their work special attention was paid to material characterization and implementation of data in the simulation model. Their focus was on minimizing the size of adhesive pockets, as they prevent interlocking. Balawender et al. [84] dealt discussion of technological aspects and experimental investigations of clinched lap joints of different metal strips combined with their adhesive bonding. Their main purpose was to analyze the influences on the geometry and mechanical strength of the clinched joint using different types of sheet metals. The experiments with application of a particular commercial software, based on the principle of digital image correlation, allowed for exact monitoring of the deformation process of the considered hybrid joint.

### 3.2. Welding Processes and Adhesive Bonding

A brief discussion on the applications of structural adhesives in the most common hybrids joining technology combining adhesive bonding and welding processes (RSW-AB and FSW-AB) is presented in this section.

#### 3.2.1. Resistance Spot Welding and Adhesive Bonding (RSW-AB)

Contemporaneous presence of a spot weld and adhesive layer leads to competitive advantages over both simple adhesive bonding and RSW. As stated by Darwish et al. [85] in different works [86,87], compared to the former, RSW-AB provides higher strength and stiffness, superior resistance to normal/peel loads and ease of manufacture. Charbonnet et al. [88] utilized three kinds of zinc-coated mild steels, a single-part epoxy adhesive and a single-part rubber sealer to experimentally demonstrate that the weld-bonded parts behave in a similar manner to continuous adhesive joints, thereby increasing their overall performance when compared to conventional spot-welded joints. The work has furthermore proved that already existing spot-welding machines can be utilized for weld-bonding applications. 

Peroni et al. [89] studied the progressive collapse behavior of some thin-walled closed-section structural sections made from deep-drawing steels and joined with different joining systems. Solutions characterized by different continuous joining technologies, nonconventional for automotive constructions, are examined and compared to the usual spot-welding solution. Different types of adhesives (acrylic and epoxy) and laser welding were considered. The authors found that continuously joined structures are at least equivalent to and generally better than spot-welded structures, and have further advantages typical of these joining solutions (higher stiffness and fatigue strength, improved vibration response, especially in the case of adhesive joints)

Sadowski et al. [90], conducted comparative experiments on spot welding joints, single-lap-bonded joints and spot-welding–adhesive joints, and the results indicated that there is an improvement in shear strength with the use of hybrid joints (see Figure 8).

Higher mechanical strength of the hybrid joints (polyurethane adhesive + spot-welded joints) compared with adhesive joints or resistance spot welded joints has been obtained in the work of Piwowarczyk et al. [91]. Similarly, Pizzorni et al. [92] conducted an experimental study on hybrid RSW-AB joints made of DP 1000 steel substrates. A semi-structural, flexible epoxy–polyurethane adhesive was used to assemble the joints. The static and fatigue were evaluated to compare the spot-welded only to adhesively bonded joints. The results obtained confirmed the reliability of the hybrid joining using RSW and elastic adhesives.

The failure process of weld-bonded joints is still not fully understood, and established failure criteria do not exist, mainly because of the coexistence of the weld nugget and the adhesive layer, which makes the stress and strain analyses more complex [93]. Campilho et al. [94] carried out an experimental and numerical study on hybrid weld-bonded single-lap unions, in comparison with the spot-welded and adhesively bonded equivalents, considering a ductile adhesive. A parametric study on the overlap length allowed proper characterization of the strength advantages of this hybrid technique under different conditions. The Finite Element Method (FEM) and Cohesive Zone Models (CZM) for damage growth were also tested to evaluate this technique for strength prediction, showing accurate estimations for all kinds of joints. To summarize, numerous studies indicate that the lap-shear strength, corrosion resistance and fatigue performance of weld-bonding joints are higher than that of traditional RSW joints [95,96,97,98]. However, weld bonding is mostly being used to gain higher static strength and stiffness, and superior fatigue strength.

#### 3.2.2. Friction Stir Welding and Adhesive Bonding (FSW-AB)

For the FSW-AB technique the use of suitable polymeric adhesives with high thermal resistance, good wetting and flow characteristics is mandatory in order to obtain a good-quality bond of the metal surfaces. The best candidates for this purpose appear to be epoxy-based adhesives (mono- or bi-component) and polyimide-based adhesives [40]. By combining FSW and adhesive bonding into a friction stir (FS) weld bonding, FSW-AB, the overall joint mechanical performance is greatly improved as the adhesive layer reduces the peel stress at the weld edges [99]. For instance, this method was used to improve the strength of FSW joints of magnesium-to-aluminum FS spot welded joints, resulting in increased quasistatic and fatigue strength [100]. Similarly, this method was proposed for continuous overlap joints of AA2024-T3, resulting in improved quasistatic and fatigue strength [101]. Lertora et al. [102] in 2019 studied FSW-AB in weld-through and flow-in configuration of AA6082 aluminum alloy, showing that the latter improved joint strength and joint fatigue life. In another study, Fortunato et al. [103] found the application of ultrasonic testing to friction stir weld-bonded joint inspection a successful method to evaluate joint quality. An innovative FSW-AB technique was investigated in the same way by Maggiore et al. [40] in 2020. They characterized an epoxy resin-based adhesive used in the hybrid FSW-AB technology which involves the use of polymeric adhesives to join metallic elements giving significant advantages as well as uniform distribution of bond–joint stress, protection from corrosion and reduced noise and vibration. The mechanical properties investigated (tensile strength and displacement) show increased values for the samples obtained via FSW-AB techniques thanks to the combined effect of the FSW and the presence of the adhesive. Experimental force–displacement curves for FSW and FSW-AB joints can be seen in Figure 9.

## 4. Conclusions and Future Perspectives

Nowadays, joining is considered a fundamental part of innovative and sustainable manufacturing. The choice of the best joining solution for a given application needs to be based on several factors, such as the desired production volume, availability of equipment and required joint performance and not ultimately the financial resources. Emerging trends in manufacturing such as light weighting, increased use of multi-material structures and the need for joining of dissimilar materials motivate growing research efforts in the hybrid joining technologies. In this review, the vast range of current and emerging hybrid joining technologies are presented. The contribution of the adhesive for each hybrid joining techniques is highlighted and main conclusions are drawn:A wide variety of adhesives are available from a range of adhesive manufacturers. The adhesive selection process is difficult as there is no universal adhesive that will fulfil every application. Properties of adhesive can vary greatly, and an appropriate selection is necessary depending on the hybrid process used. Epoxy adhesives use is still the most widespread, due to their peculiar characteristics (high strength, temperature resistance, low cure temperatures, ease of use, and low cost). The polyurethane adhesives are mainly used in hybrid joints because of their superior flexibility at low temperatures, resistance to fatigue, impact resistance and durability. Polyimide-based adhesives find their niche in the case of the hybrid joining technology FSW-AB, due to the need to use suitable polymeric adhesives with high thermal resistance, good wetting and flow characteristics in order to obtain a good-quality bond of the metal surfaces;In a hybrid-bonded fastened joint the load sharing depends on a combination of materials and geometric parameters. Moreover, as confirmed by several authors, hybrid joints using rivets or bolts bear an increased maximum load and have higher initial stiffness, strength and energy absorption in comparison with nonhybrid joints. The improvement in hybrid fastened-bonded joints performance compared to both individual joints is achievable when there is substantial load sharing between the adhesive and the fastener. The load sharing in these joints depends on a combination of materials and geometric parameters and attaining appropriate load sharing in the hybrid joint design is a challenge. Most of the new advanced fastened hybrid joining processes (e.g., mechanical clinching and self-pierce riveting) do not need an additional connecting part, which helps to reduce weight. They also do not require predrilled holes. Thus, they enable a sensible reduction of the overall joining time;Adhesive bonding combined with welding technologies provides joints with improved fatigue characteristics when compared to spot welding, due to reduction in stress concentrations at the weld-nugget periphery. They also exhibit improved mechanical and thermal properties. In particular, in weld-bonded joints the stress concentration decreases providing better mechanical properties as an increase in tensile shear and/or compressive buckling load. Finally, another key feature of the weld-bonding technology is the optimization of the quality of the sealing and the improvement of the overall resistance to corrosion. However, some technologies are not currently ready for practical application in industry (e.g., combining adhesive bonding and arc or beam fusion processes) and further developments are necessary.

Some critical challenges and future research directions are summarized below:Due to their complexity and relatively recent development, there is a need for the development of robust design and process simulation tools for hybrid joining processes. The development of numerical models of the hybrid processes would allow a better understanding of all the phenomena involved. At the same time, they will represent a valuable tool for determining the influence of process parameters on the quality of the hybrid joints. This would greatly advance the understanding of these processes and promote further process development and optimization to expand their area of applications in industry;There is a need to investigate the effect of environmental factors (e.g., the influence of radiation, humidity and temperature) on the hybrid joints. These issues will need to be addressed before any potential implementation of hybrid joining technologies involving adhesives in industry;The development of new adhesives is continuously increasing. The adhesives need to have some specific characteristics depending on the hybrid process used. For example, for fastening adhesive hybrid joints the ideal adhesive to be used should have low stiffness, extreme elongation to failure and reversible deformation, while for weld-bonded joints the adhesives need to withstand high temperatures. There is a need to study a new class of adhesive (e.g., elastomers) to be used in hybrid joining processes;In several industries (e.g., aerospace, automotive, rail and naval transport industries), the use of hybrid joining technology could be considered as a potential solution in order to reduce the mass as well as the manufacturing cost. In addition, due to the increasing aspirations for more environmentally friendly technologies and lightweight materials, the joining of polymers, composites and multi-material hybrid structures for industrial applications is still a growing research and development area. Thus, there is a need for new developments on advanced hybrid joining technologies for new advanced materials.

## Figures and Tables

**Figure 1 polymers-13-03961-f001:**
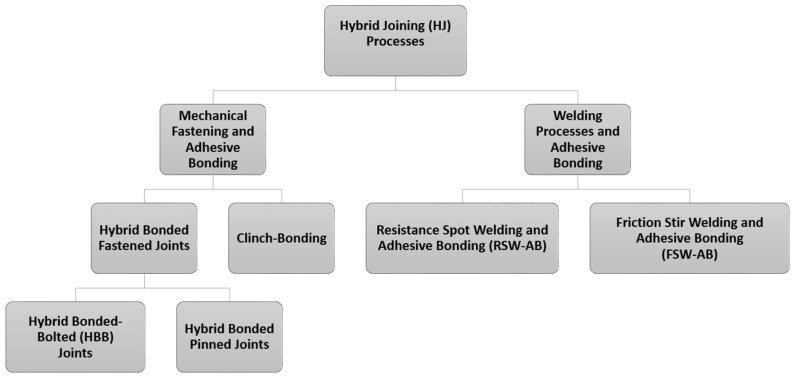
Graphical scheme of the structure of the techniques selected and described in the present review.

**Figure 2 polymers-13-03961-f002:**
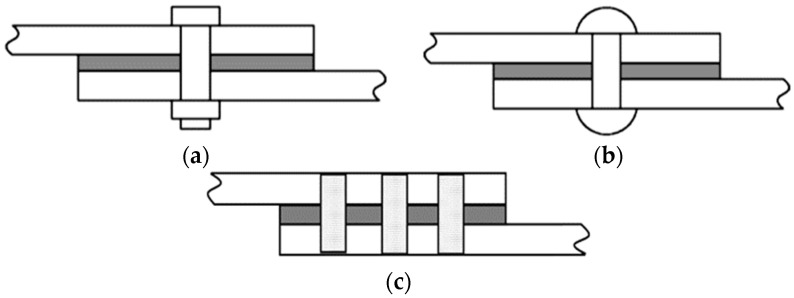
Schematic representations of (**a**) bolted-bonded joint, (**b**) rivet-bonded joint and (**c**) a pin-bonded joint.

**Figure 3 polymers-13-03961-f003:**
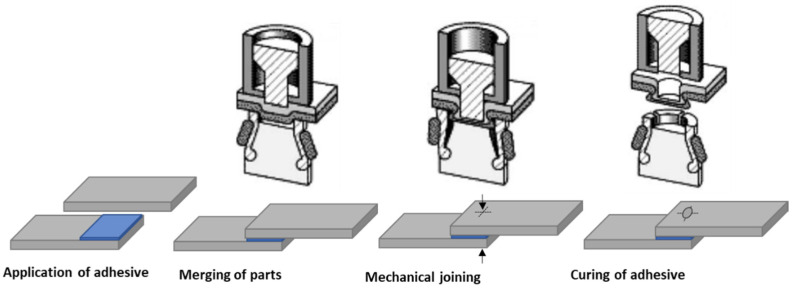
The principle of the clinch-bonding technology based on [32].

**Figure 4 polymers-13-03961-f004:**
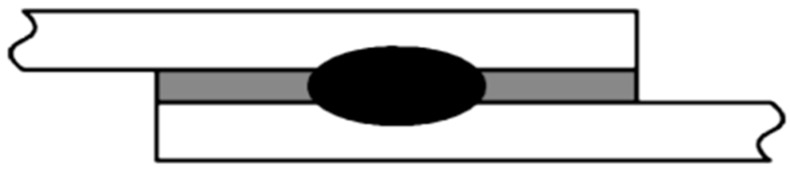
Weld-bonded joints result from the combination of adhesive bonding with resistance spot-welding based on [23].

**Figure 5 polymers-13-03961-f005:**
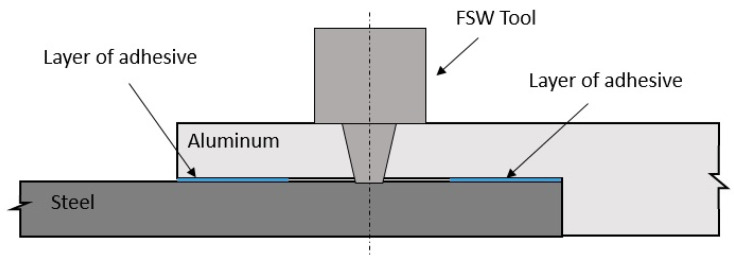
Schematic illustration of the hybrid joining technology FSW-AB [40].

**Figure 6 polymers-13-03961-f006:**
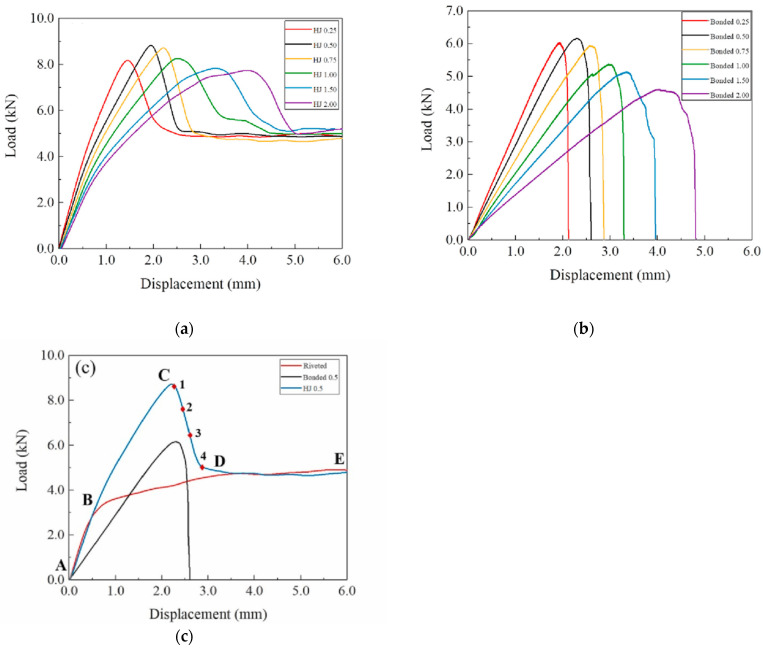
Load-displacement curves of HJs (**a**), bonded joints (**b**) and (**c**) the comparison of three types of joints [63].

**Figure 7 polymers-13-03961-f007:**
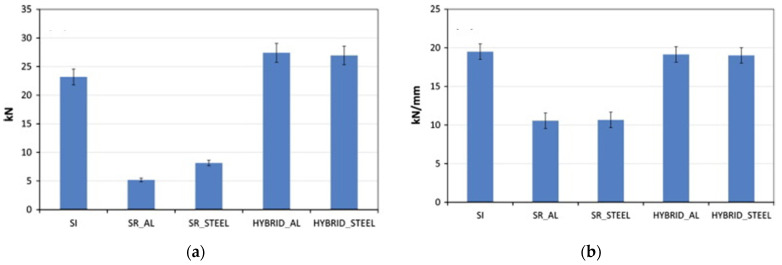
(**a**) Average tensile failure load and (**b**) average stiffness of the joints analyzed. Numerical experimental analysis of hybrid double lap aluminum-CFRP joints [75].

**Figure 8 polymers-13-03961-f008:**
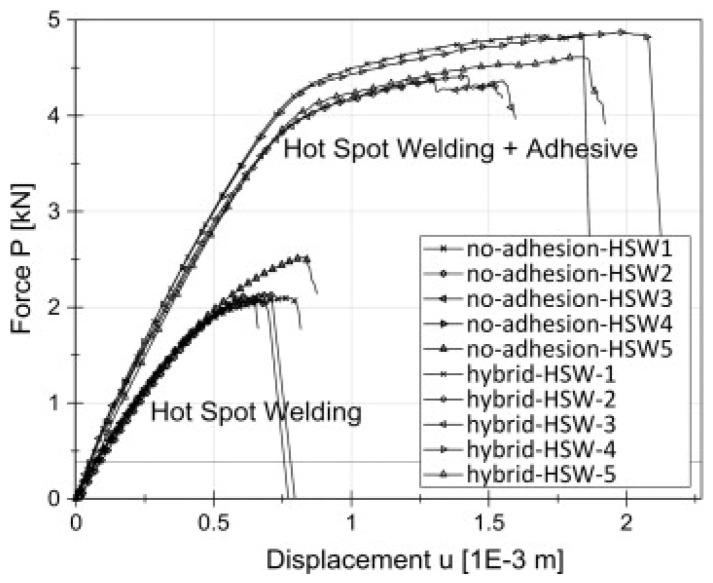
Experimental force–displacement diagrams of the spot weld joints and the hybrid joints (combining spot welding and adhesive bonding) [90].

**Figure 9 polymers-13-03961-f009:**
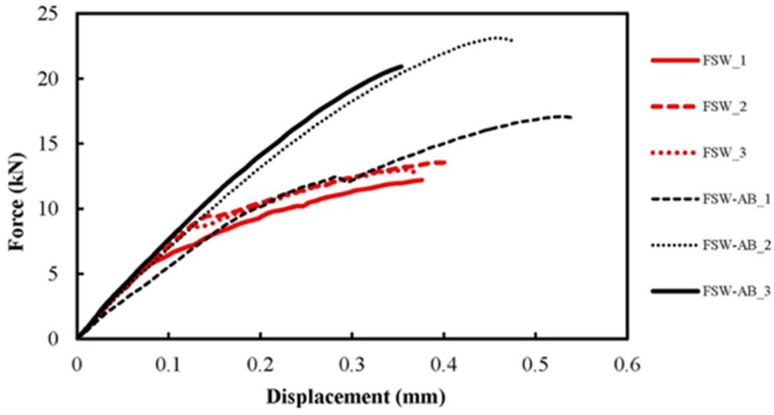
Force–displacement curves of FSW and FSW-AB joints [40].

**Table 1 polymers-13-03961-t001:** Process parameters of hybrid bonded fastened joints [24,26].

Parameters of Workpiece Materials	Geometric Parameters	Type of Joint Configuration	Adhesive	Types of Fasteners ^1^
Nature of material	Thickness of adherends	Single-lap and double-lap joints	Epoxy	Sphere type pin fastener
Thickness	Length of overlap	Stepped lap joint	Polyurethane	Wedge type pin fastener
Similar or dissimilar system of materials	Adhesive Thickness	Scarf joint		Rounded shaped head type fastener
Strength of material	Fastener–hole clearance	T-joint		Protruding type fastener head
Young’s modulus of material	Width	Butt joint		Countersunk type fastener head

^1^ Fasteners: pin mean diameter less than 1 mm without top head, bolt and rivet are denoted as fasteners.

## Data Availability

Not applicable.

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
