# Peer review of "A Review of Structural Adhesive Joints in Hybrid Joining Processes"

_polymers, 2021, doi:10.3390/polym13223961_

Round 1

Reviewer 1 Report

The review manuscript 1448808 describes about hybrid joining that worth practical design of systems especially for airplane and spaceplane. The manuscript seems to need to be corrected some points. My comments for the manuscript are as follows.

  1. Line211, Figure 2: The figure is only about clinching not hybrid. We readers want to see a clinch-adhesive joints (Hybrid) of reference 33, because this review is about hybrid process.
  2. Line233-239: Authors mentioned about RSW-AB by using the Figure 3. The welded bedad will include some contamination from the adhesive agent which have carbon etc.. Do their impurities have a negative effect for strength of the welding joint? 
  3. Line 265‐281,Figure 4: I have same comment about Figure 4. FSW-AB bead include something impurities from the adhesive agent. Are there something negative effect or not?
  4. Figure 5.: The data of Figure 5(a) are all saturated to 5 kN. What does this mean? Is the adhesive bond broken and the rivets remain and the load rise to yielding stress?
  5. Figure 5(a),(b) : What does the meaning of numerical values 0.25-2.00 in these figures? I want know because the numerical values have big effect on each result?
  6. Line 400-412, Figure6.(a): Why the effect of Hybrid rivet-bonded joint is small for average tensile failure load as shown in Figure 6?
  7. Line 485-487: Why the effect of hybrid spot welding and adhesive is so big as shown in Figure 7?
  8. Line 412: I want to know about the result of fatigue performances, more. Because the fatigue clack propagation from rivet possible to lead to a big accident of airplane.

There is a possibility to prevent the fatigue crack propagation by hybrid process.

That's my comments.

Author Response

Dear Editor and Reviewers,

The authors would like to thank for all the comments and considerations, which allowed the authors to improve the quality of the manuscript. Changes made according to the reviewers’ suggestions are marked in ‘red’ (we apologize, but the review system of word did not let us to make all the changes without compromising the final readability, if there are problems please let us know and we will try to fix whatever possible for highlight of the changes). We hope that all the suggestions have been attended and that the paper is now suitable for publication.

The comments and response are also included below.

Reviewer 1

  1. Line211, Figure 2: The figure is only about clinching not hybrid. We readers want to see a clinch-adhesive joints (Hybrid) of reference 33, because this review is about hybrid process.

Response: The authors are thankful to the Reviewer for kind suggestions, a figure was added to better explain the process.

  1. Line233-239: Authors mentioned about RSW-AB by using the Figure 3. The welded bedad will include some contamination from the adhesive agent which have carbon etc.. Do their impurities have a negative effect for strength of the welding joint? 

Response: See response below

  1. Line 265‐281,Figure 4: I have same comment about Figure 4. FSW-AB bead include something impurities from the adhesive agent. Are there something negative effect or not?

Response: We search in literature for this specific topic. The focus of the inclusion of the adhesive, as far as the authors were able to search and find, is mainly on the deterioration of the mechanical performance while there is no studies about the eventual effect of the impurities due to the degradation of the adhesive involved in the joining process. Indeed, the presence of inclusion should degrade the anticorrosion performance even if it is also to consider the sealant action of the adhesive. This is also the topic of our research, preliminary results for our specific case (FSW-AB) did not show (after aging in climatic chamber) any difference but the study is still ongoing

  1. Figure 5.The data of Figure 5(a) are all saturated to 5 kN. What does this mean? Is the adhesive bond broken and the rivets remain and the load rise to yielding stress?

Response: A sentence was added to explain the results presented

  1. Figure 5(a),(b) : What does the meaning of numerical values 0.25-2.00 in these figures? I want know because the numerical values have big effect on each result?

Response: The numerical values reported in the legend of the plot in figure 5 (a) (b) referrers to the thickness (in mm) of five kinds of bond line reported from Zhang et al [ref 69]

  1. Line 400-412, Figure6.(a): Why the effect of Hybrid rivet-bonded joint is small for average tensile failure load as shown in Figure 6?

Response: See response for question 7

  1. Line 485-487: Why the effect of hybrid spot welding and adhesive is so big as shown in Figure 7?

Response: In order to help the reader to better compare the different rivet bonded joints we added a sentence (lines 449-451) with additional info.

  1. Line 412: I want to know about the result of fatigue performances, more. Because the fatigue clack propagation from rivet possible to lead to a big accident of airplane. There is a possibility to prevent the fatigue crack propagation by hybrid process.

Response: A sentence and a reference was included in the manuscript.

Reviewer 2 Report

I read carefully the review article entitled ‘A review of structural adhesive joint in hybrid joining processes for reputed Polymers. This review article is comprehensive, fits and suitable to publish in Polymers. This manuscript is generally well written and clearly presented however still need to be addressed many comments and thus require moderate revision before its acceptance.

  • Title need to modify which can describe whole research work.
  • Abstract should be rewrite add more detail of results and importance of the study.
  • A well addressed graphical scheme of study design should be inserted.  
  • In the introduction section, write the novelty of the work and the problem statement clearly.
  • More discussion about the structural adhesive joint their current production, types of materials used and why this review article is important.
  • The major lacking part of this review article authors have described everything in the text format. It looks like essay. Instead of make one or two tables describing the review of literature according to their applications is highly recommended.
  • Few figures by referring the literature about the various applications is essential to make this review article more interesting.
  • The conclusion of the study is not discussed with the specific output obtained from the study, it could be modified with precise outcomes with a take home message.
  •  English and grammar mistakes are present author should check the manuscript carefully to improve the quality of the manuscript.

Author Response

Dear Editor and Reviewers,

The authors would like to thank for all the comments and considerations, which allowed the authors to improve the quality of the manuscript. Changes made according to the reviewers’ suggestions are marked in ‘red’ (we apologize, but the review system of word did not let us to make all the changes without compromising the final readability, if there are problems please let us know and we will try to fix whatever possible for highlight of the changes). We hope that all the suggestions have been attended and that the paper is now suitable for publication.

The comments and response are also included below.

Reviewer 2

  1. Title need to modify which can describe whole research work.

Response: In the author’s opinion the title is adequate with the manuscript content.

  1. Abstract should be rewrite add more detail of results and importance of the study.

Response: The abstract was totally rewritten and improved.

  1. A well addressed graphical scheme of study design should be inserted.  

Response: We added at the beginning of the work a graphical scheme of the structure of the techniques selected and described in the review.

  1. In the introduction section, write the novelty of the work and the problem statement clearly.

Response: The novelty of the work was included in the introduction. See “There are many recent review articles regarding joining techniques of same and dissimilar materials combining mechanical fastening, welding and the use of adhesives The focus of theses reviews is to describe the process of joining and their mechanical and physical properties via simulation and laboratory experiments [1-8]”

  1. More discussion about the structural adhesive joint their current production, types of materials used and why this review article is important.

Response: In the review we choose to present the adhesives on their chemical natures, we added why we believe that this review can be a contribution for the reader of Polymers.

  1. The major lacking part of this review article authors have described everything in the text format. It looks like essay. Instead of make one or two tables describing the review of literature according to their applications is highly recommended.

Response: we added one table to add as a S1.

  1. Few figures by referring the literature about the various applications is essential to make this review article more interesting.

Response: We thank the reviewer for the suggestion. We added one more figure. Since the paper include already eight figures + two tables in order to conform to the journal we considered the number of figures sufficient.

  1. The conclusion of the study is not discussed with the specific output obtained from the study, it could be modified with precise outcomes with a take home message.

Response: The conclusions were improved according to the reviewer’s suggestions.

English and grammar mistakes are present author should check the manuscript carefully to improve the quality of the manuscript.

The language of the article was rechecked for English and grammar mistakes and improved from the previous version

Round 2

Reviewer 2 Report

The authors have substantially revised the manuscript according to the comments.

The present form of the manuscript can be accepted for publication.